# communications
# engineering

# Effect of thermal gradients on inhomogeneous degradation in lithium-ion batteries

Shen Li[1,2,5], Cheng Zhang[3], Yan Zhao[4], Gregory J. Offer [1,2] & Monica Marinescu [1,2✉]

Understanding lithium-ion battery degradation is critical to unlocking their full potential. Poor understanding leads to reduced energy and power density due to over-engineering, or conversely to increased safety risks and failure rates. Thermal management is necessary for all large battery packs, yet experimental studies have shown that the effect of thermal management on degradation is not understood sufficiently. Here we investigated the effect of thermal gradients on inhomogeneous degradation using a validated three-dimensional electro-thermal-degradation model. We have reproduced the effect of thermal gradients on degradation by running a distributed model over hundreds of cycles within hours and reproduced the positive feedback mechanism responsible for the accelerated rate of degradation. Thermal gradients of just 3 °C within the active region of a cell produced sufficient positive feedback to accelerate battery degradation by 300%. Here we show that the effects of inhomogeneous temperature and currents on degradation cannot and should not be ignored. Most attempts to reproduce realistic cell level degradation based upon a lumped model (i.e. no thermal gradients) have suffered from significant overfitting, leading to incorrect conclusions on the rate of degradation.

[1] Department of Mechanical Engineering, Imperial College London, London SW7 2AZ, UK. [2] The Faraday Institution, Harwell Science and Innovation Campus, Didcot OX11 0RA, UK. [3] Institute for Future Transport and Cities, Coventry University, Coventry CV1 5FB, UK. [4] Breathe Battery Technologies Limited, London SE1 7SJ, UK. [5] Present address: Rimac Technology R&D UK Limited, Warwick CV35 9EF, UK. ✉email: monica.marinescu@imperial.ac.uk

Lithium-ion batteries (LIB) are essential to the future of the transportation and electricity generation sectors. However, batteries degrade over time, limiting their lifetime and increasing the risk of catastrophic failure of the energy storage system. How fast degradation occurs remains a critical uncertainty for many applications[1].

The degradation of LIBs is influenced by many factors, such as the battery chemistry and material[2], manufacturing process, and the operating conditions, including the temperature[3–5], current rate[6–8] and state of charge (SoC)[2,9,10]. Understanding stress factors is vital to predicting the battery's service life, which, in turn, is important for battery warranty and maintenance such as replacing/retiring schemes. Predicting battery life is also key to optimising the system design and real-time control of the battery pack, such as derating the charging/discharging power as the battery ages to prevent safety hazards.

It was found experimentally that inhomogeneous intra-cell degradation is common for pouch cells[11,12], cylindrical cells[13,14] and coin cells[15]. Thermal management was found to play an important role in slowing down degradation processes for single-cell and battery packs due to inhomogeneities[16–18]. However, it was shown experimentally that different methods of actively cooling a pouch cell can affect its rate of degradation significantly, up to three times for a high-power pouch cell[19], contradicting the prevailing wisdom at the time, that a cell at a higher temperature would undergo faster degradation. This finding was hypothesised to be caused by non-uniform temperature distributions inside the cell, leading to gradients in resistance and current, and in turn to inhomogeneities in local degradation within the cell. The thermal inhomogeneities inside the cell were confirmed in a later publication by a two-dimensional distributed electro-thermal model for beginning of life (BoL) simulations[20]. However, with a BoL model only, the effects of temperature gradients on degradation could still only be hypothesised, rather than demonstrated. To date, there remains a crucial missing link between thermal gradients and degradation.

Considering that temperature inhomogeneities have been demonstrated experimentally to enhance the rate of degradation[19,21], it can be argued that most models assuming homogeneous degradation are likely to have overestimated the rate of degradation, possibly with significant consequences. Thermally lumped degradation models[22–24] fundamentally lack the ability to reproduce the interaction between the cooling strategy, or any application-relevant thermal boundary conditions, and the degree of degradation inhomogeneity. Thus lumped models cannot correctly predict the acceleration in overall cell degradation. Modelling degradation over many cycles necessarily carries a significant computational cost for three-dimensional physics-based models, which is a possible reason why studies investigating these effects are lacking in the literature. In order to reduce complexity, models are normally used either without a thermal component[4,25], or with a single temperature and state of health (SoH) for the entire cell[26,27].

To predict the effect of temperature gradients on the rate of degradation, a thermally coupled discretised cell model is required, where temperature, current density and SoH are local states that can vary within the cell. The two most commonly used types of models for degradation are continuum models and equivalent circuit models. Continuum models can describe various important ageing mechanisms, such as solid-electrolyte interphase (SEI) layer growth[28], Li plating[29] and electrode cracking[30], with equations describing the physical principles of these degradation mechanisms. These models are usually based on the either the Doyle-Fuller-Newman (DFN) model[31], or the pseudo-two-dimensional model (P2D)[32], or on the single particle model (SPM)[33]. Although multi-scale approaches[34,35] have been created based on the P2D model to facilitate the modelling of inhomogeneities at cell level, these are not suitable for simulating battery degradation over many hundreds or thousands of cycles due to their computational cost. In contrast, empirical models[36,37] or semi-empirical models[38,39] use relatively simple mathematical functions for capacity loss and resistance increase. So far, these models have been oversimplified either in space[36] or time: long-term cycling simulation is avoided by fitting cycling ageing parameters[38,39] or the degradation history of hundreds of cycles is approximated by one ageing step[37]. Therefore they do not accurately describe the cycle-by-cycle evolution and interaction between thermal, electrical and degradation inhomogeneities.

This work proposes a novel distributed electrical-thermal-degradation equivalent circuit network (ECN) model with the ability to track the cell's local states as they evolve cycle-by-cycle. A degradation function for capacity and power fade is adapted from previously published studies and used within the 3D distributed electro-thermal model to progress degradation, in the form of increased resistance, incrementally and locally within the cell. This allows the model to retrieve the interactions between the inhomogeneities in temperature, resistance, current, state of health and rate of degradation. The effects of surface versus tab cooling on cell degradation, as experimentally observed by Hunt et al.[19], are retrieved and interpreted via access to the model states. The mechanisms underpinning the effect of thermal management on the degradation rate of the cell are revealed by analysing the evolution of the distributed states within the cell during the full ageing history. To provide supplementary understanding, the complex degradation processes are analysed by considering a simplified pseudo end-of-life (pEoL) model, used to efficiently explore the effects of resistance inhomogeneity on the available cell capacity.

## Methods
The electro-thermal-degradation model used in this study is introduced as follows: electro-thermal model setup (section "The electro-thermal ECN model") and validation (section "Validation of electro-thermal ECN model"); the electro-thermal-degradation model for long-term simulation (section "Electrical-thermal-degradation model"); the simple pEoL model (section "A pseudo end-of-life (pEoL) model") created to explore qualitatively the effect of resistance and current inhomogeneity while avoiding the computational cost of long-term cycling ageing.

**The electro-thermal ECN model.** An electro-thermal model for a pouch cell was developed using the Finite Difference Method, as previously used in the creation of a 3D cylindrical cell model[40] and a 2D pouch cell model[20]. Figure 1 illustrates the 3D distributed thermally coupled ECN. The domain of the computational unit is composed of an electrode pair subdomain (i.e., anode, separator and cathode) and two current collector subdomains. Each electrode pair and current collector subdomain along the length ($x$-axis), width ($y$-axis) and thickness directions ($z$-axis) is represented by an electrical and a thermal ECN unit, two-way coupled. Here a distributed model of 45 electrical/thermal ECN units is used: 3 units along $x$-axis, 3 units along $y$-axis and 5 units along $z$-axis. There are 100 electrode pair layers as measured in the disassembly test[20]. For computational efficiency, 5 units along $z$-axis are chosen with each ECN unit along the $z$-axis representing 20 layers. A convergence check was conducted to verify the accuracy of this level of discretisation. For more details, please see the Supplementary Note 1).

For each electrode pair subdomain, the current and voltage are related via an ECN, which contains a voltage source $E_s$, a series resistance $R_0$ and a set of Resistor-Capacitor (RC) branches. Within each electrical ECN unit, three RC branches are used, in order to retrieve the dynamic response of the cell for a wide range of time constants. The terminal voltage $\Delta\phi^{El}$ of each ECN unit is

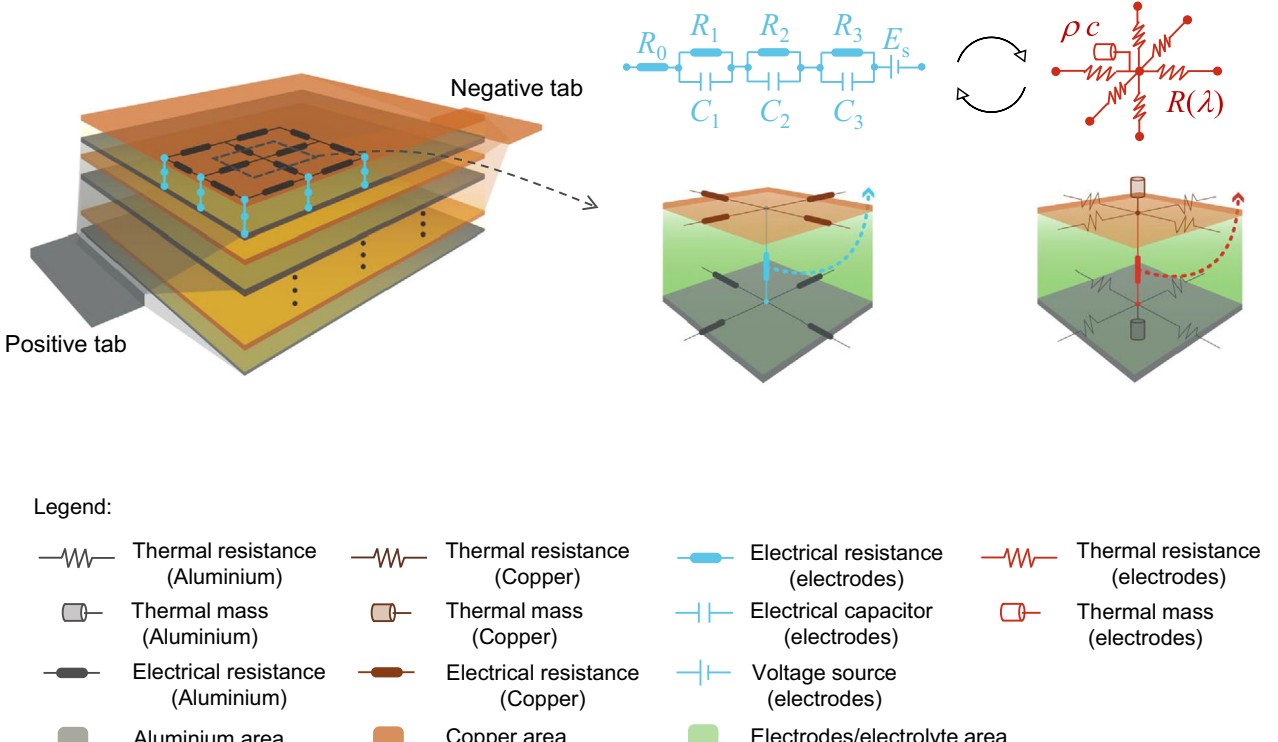

**Fig. 1 Schematic representation of the electro-thermal model for a pouch cell.** The grey and orange regions represent the computational domain of aluminium and copper current collectors, respectively. The blue and red networks constitute the electrical and thermal models for each electrode pair subdomain.

then given by:

$$\Delta\phi^{El} = E_s - \sum_{i=1}^{3} R_i I_i - R_0 I, \quad (1)$$

where $i$ is the index of the RC branch, $I_i$ is the branch current through $R_i$, and $I$ is the current through $R_0$. The resistances, capacitances and the open circuit voltages (OCV) of each electrical ECN unit are functions of state of charge (SoC) and temperature. For the current collector domain, the charge balance is:

$$\sigma\nabla \cdot (\nabla\phi^{CC}) = \sigma\frac{\partial^2\phi^{CC}}{\partial x^2} + \sigma\frac{\partial^2\phi^{CC}}{\partial y^2} = 0, \quad (2)$$

where $\sigma$ is the conductivity of the current collector (aluminium foil for the positive electrode and copper foil for the negative electrode) and $\phi^{CC}$ is the corresponding local electric potential.

Within each thermal ECN unit (describing either a current collector or electrode pair), heat transfer along the $x$, $y$ and $z$ directions is described by the diffusion equation:

$$\rho c\frac{\partial T}{\partial t} = \lambda_x\frac{\partial^2 T}{\partial x^2} + \lambda_y\frac{\partial^2 T}{\partial y^2} + \lambda_z\frac{\partial^2 T}{\partial z^2} + q, \quad (3)$$

where $\rho$ is mass density, $c$ is specific heat capacity, $\lambda$ is heat transfer coefficient and $T$ is temperature of the material. $\lambda_x$, $\lambda_y$ and $\lambda_z$ are the heat transfer coefficients along the three axes, and $q$ is the lumped heat source within the unit volume.

The thermal model accepts multiple types of thermal boundary conditions, described below.

(i) Convection boundary condition:

$$q_{conv} = h(T - T_{amb}), \quad (4)$$

where $q_{conv}$ is the heat transferred via convection, $h$ is the convective heat transfer coefficient, $T$ is the temperature on

the boundary of the unit volume and $T_{amb}$ is the ambient temperature.

(ii) Fixed surface temperature.

(iii) Any combination of (i) & (ii).

As the various locations within the cell are represented distinctly within the model, neighbouring regions on the surface of the cell can be subjected to different boundary conditions. This enables the model to simulate various types of thermal management strategies relevant to practical battery storage systems.

The material thickness and thermal properties are taken from previous work on the same pouch cell manufactured by Kokam (Model: SLPB11543140H5)[20].

The electrical and thermal models are two-way coupled. The heat source in the electrode-pair domain is given by:

$$q^{El} = \left(\sum_{i=1}^{3} I_i^2 R_i + I^2 R_0 + IT\frac{dE_s}{dT}\right)/V_e^{El}, \quad (5)$$

where $V_e^{El}$ is the electrode pair volume in each subdomain. The overall heat source in Eq. (3) is thus composed of an irreversible term and a reversible term. The heat source $q^{CC}$ for the current collector subdomain is:

$$q^{CC} = \left(\frac{1}{2}\sum_{j=1}^{4}\left(I_j^{CC}\right)^2 R_j^{CC}\right)/V_e^{CC}, \quad (6)$$

where $I_j^{CC}$ and $R_j^{CC}$ are the current and resistance in the aluminium and copper foil domains. The local temperature affects, in turn, the values of the elementary components of the electrical ECN, i.e., OCV, R's and C's.

The parameters of the electrical model for the whole cell are formulated as lookup tables at different temperatures (10, 20, 30 and 40 °C) and different SoCs (from 1% to 100%). The electrical parameters (open circuit voltage $E_s$, resistance $R_0$, $R_j$ and capacitance $C_j$) are extracted using Layered method[41] on

experimental data from Galvanostatic Intermittent Titration Technique (GITT) tests conducted in discharge at 1 C (5 A). The electrical parameters and thermal properties (thermal conductivity $\lambda$, specific heat capacity $c$ and mass density $\rho$ for aluminium foil, copper foil, anode, cathode and separator) are taken from previous work on the same cell[20].

**Validation of electro-thermal ECN model.** The electro-thermal ECN model is validated against two series of experimental scenarios. The experimental data is published in a previous work[20] on this pouch cell. The first validation experiment is conducted using the US06 drive cycle on the cell under adiabatic conditions, in thermal chamber, with the ambient temperature set as 20 °C, with results shown in Fig. 2(a). This experiment is designed to validate the electro-thermal model under conditions when inhomogeneities are expected to be at a minimum. A convective heat transfer coefficient of $h = 6.5 W \cdot m^{-2} K^{-1}$ allows the predicted terminal voltage to match the measured cell voltage with an RMSE of 46.99 mV over the whole experiment, as shown in Fig. 2(a). The measured surface temperature at the centre of the cell surface is predicted by the model with an RMSE of 0.60 °C, as shown in Fig. 2(b). Based on these results, it is concluded that the model describes sufficiently well the coupled electrical and thermal behaviours for the cell when no significant internal inhomogeneities are expected to exist.

Two thermal management scenarios are applied to validate the model when thermal gradients are expected within the cell and to affect its performance: single-sided surface cooling (SC) and (both) tab cooling (TC). In both scenarios, the cooled surfaces are kept at 20 °C by Peltier elements, while the other surfaces are exposed to convection at 20 °C in the thermal chamber. Thermocouples are placed at the centre of the cell surface. The fully charged cell, equilibrated at 4.2 V and 20 °C, is discharged under a constant current of 6 C (30 A) to a voltage cutoff of 2.7 V, and then immediately charged with a constant current of 2 C (10 A). With no fitting parameters, the model prediction matches the experimental surface temperature reported previously[20], with an RMSE of 0.92 °C (for SC) and 1.35 °C (for TC), as shown in Fig. 2(c), (d). The mismatch between experiment and model predictions is only apparent where the cell surface temperature peaks, corresponding to SoCs below 20%. A probable reason for this mismatch is the fact that the data used for parametrisation was taken at 1 C, while the validation experiment was performed at 6 C discharge and 2 C charge. The values of the resistances in the ECN are expected to be current-dependent, but this dependence is neglected in the current work. Under 6 C constant current discharge, the simulated voltage matches the experimental results, with RMSE of 184.71 mV (for SC) and 150.82 mV (for TC), as shown in Fig. 2(e), (f). Based on the small RMSE in these results, it is concluded that the proposed model is suitable for retrieving the effects of electrical and thermal non-uniformities under the two opposing thermal management protocols.

**Electrical-thermal-degradation model.** The distributed electrical-thermal-degradation model is created to allow for time-dependent degradation inhomogeneities during cycling. In this degradation model, resistance increase and capacity loss caused by SEI growth are considered by adopting the ageing law parameterised in the study by Cordoba-Arenas et al.[42], the only experimental study that provides degradation functions with dependency on both temperature and current density over a wide range of values. The ageing law for capacity loss is written as a function of temperature and charge throughput $W$. The capacity loss percentage relative to the rated capacity $\eta$ is given by[42]:

$$\eta = 5.57 W^{0.48} \exp\left(-\frac{2694.97}{T}\right), \tag{7}$$

where $T$ is the surface-controlled temperature during cycling. In the distributed model developed in this work, this capacity loss is applied in a distributed manner to the multiple cell units, as a function of the local charge throughput density $w$, defined as:

$$w = \frac{W}{V_0}, \tag{8}$$

where $V_0$ is the total volume of the active material undergoing degradation, i.e., of the electrode pair (i.e., anode, separator and cathode). For the cycle-by-cycle analysis, the differential of the ageing law in Eq. (7) is used:

$$\frac{d\eta_i}{dw} = 2.67 V_0^{0.48} w^{-0.52} \exp\left(-\frac{2694.97}{T}\right), \tag{9}$$

with $\frac{d\eta_i}{dw}$ the incremental capacity loss percentage for a unit of charge throughput density through the $i$-th ECN unit.

In the distributed model, the capacity $Q_i$ for the $i$-th ECN unit is:

$$Q_i = Q_i^{BoL} \cdot \left(1 - \frac{\eta_i}{100}\right), \tag{10}$$

where $Q_i^{BoL}$ is the begin-of-life capacity for $i$-th ECN unit. The total resistance percentage increase relative to the BoL total resistance for the cell is expressed as in Cordoba-Arenas et al.[42] relative to the total charge throughput through the cell $W$:

$$\mu = \left\{3205.3 + 36.34 \cdot \exp[0.92(5 - CR)]\right\} \cdot \exp\left(\frac{-51800}{8.31T}\right) \cdot W, \tag{11}$$

where $CR$ is the C rate at which the cell is cycled. The resistance percentage increase per volumetric density of charge throughput ($w$) in the $i$-th ECN unit corresponds to:

$$\frac{d\mu_i}{dw} = \left\{3205.30 + 36.34 \cdot \exp[0.92(5 - CR)]\right\} \cdot \exp\left(\frac{-51800}{8.32T}\right) \cdot V_0. \tag{12}$$

It is assumed that an increase in local total resistance as a result of degradation occurs as a result of proportional increases in each of the four resistances of the ECN unit. A degradation-caused increase $dR_i$ for a resistance in the $i$-th ECN unit at time $t$ is:

$$dR_i(t) = R_i^{BoL}(SoC_i(t), T_i(t)) \cdot \frac{1}{100} d\mu_i(t), \tag{13}$$

where $SoC_i(t)$ and $T_i(t)$ are the state of charge and temperature for the $i$-th ECN unit at time $t$. Thus the resistance increase $dR_i$ is affected by SoC and temperature indirectly, through $R_i^{BoL}(t)$, and by C rate directly through $d\mu_i(t)$. Any of the four resistances in the $i$-th unit ECN $R_i$ are assumed to be composed of BoL resistance plus a history-dependent increase, expressed as:

$$R_i(t) = R_i^{BoL}(t) + \int_{\tau=0}^{t} dR_i^{BoL}(\tau). \tag{14}$$

The capacity loss and resistance increase are added into the electrical-thermal ECN model, based on the modified ageing law, as detailed in Supplementary Note 2.

The model with distributed degradation is used for two common cooling strategies for pouch cells, SC and TC. For SC, the temperature of the top and bottom surfaces is fixed at 20 °C, while other surfaces are thermally insulated, i.e., are assumed to have zero heat transfer. For TC, the temperatures of the current collector nodes at the positive and negative terminals are fixed at

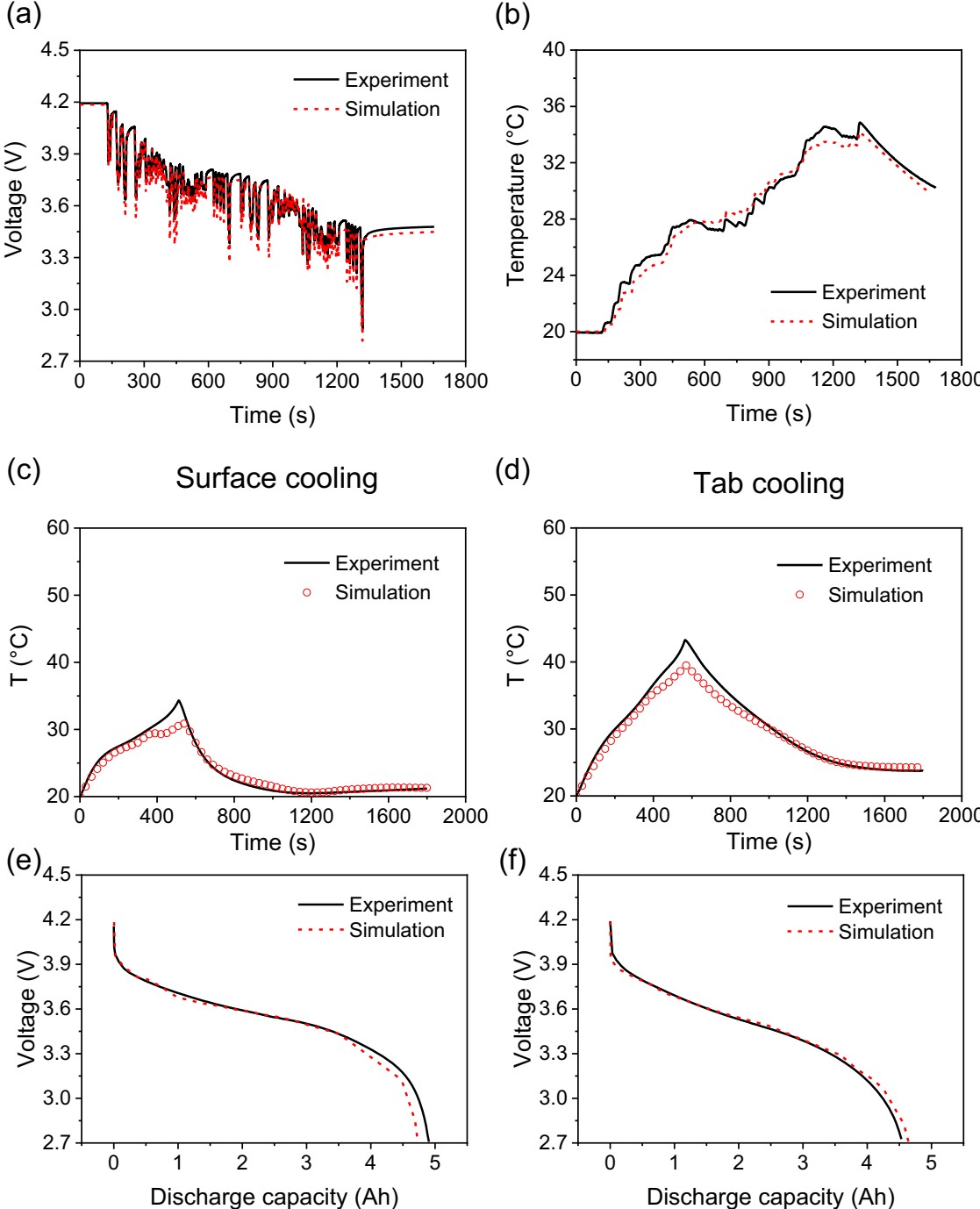

**Fig. 2 Electro-thermal model validation. Under adiabatic condition. a** Drive cycle voltage comparison, **b** surface temperature comparison. Under 6C constant current discharge, the temperature at the centre of the measured surface for **c** single-sided surface cooling and **d** tab cooling; cell voltage for **e** single-sided surface cooling and **f** tab cooling. Experimental data is obtained from previous work[21].

20 °C, while other surface nodes are thermally insulated, i.e., have zero heat transfer.

**A pseudo end-of-life (pEoL) model**. Ageing inhomogeneities emerge often during cell operation[11,12] and can be characterised by resistance inhomogeneities. Because long-term cycling ageing is a complex and computationally costly process, the simple pEoL model is created to explore qualitatively the effect of resistance inhomogeneity on the available cell capacity, while keeping the computational cost to a minimum. The pEoL could, in the

future, also serve as the basis for state estimation and degradation diagnostics.

In the pEoL model, the BoL resistance in each unit is multiplied by a coefficient $k_i$, which evolves linearly across the thickness of the cell, i.e., along the $z$-axis in Fig. 1, with $k$ denoting the unit number along this direction. The coefficients $k_i$ increase linearly from a minimum at the unit at one end of the cell to a maximum for the unit cell at the opposite end. Therefore, the modified resistance $R_k^{EoL}$ in the pEoL model is given by:

$$R_i^{EoL} = R_i^{BoL} \cdot k_i, \tag{15}$$

where $R_i^{BoL}$ and $R_i^{EoL}$ are the begin-of-life and end-of-life resistances, respectively, functions of SoC and temperature.

## Results and discussion

The long-term cycling degradation of lithium-ion battery is a complex process because it involves the interplay of electrochemical, thermal and degradation variables, e.g., resistance increase and capacity loss. In the section "The coupled effects of inhomogeneous current distribution and local resistance on available cell capacity", a qualitative study is conducted on the effects of resistance inhomogeneity on available cell capacity and the interaction between inhomogeneities of resistance and current. Section "Choosing a degradation function with current dependency" discusses common degradation functions that are necessary to facilitate this interaction. The study demonstrates the effect of simple inhomogeneities on cell performance, prior to an analysis of the full model predictions for long-term cycling degradation under thermal management, presented in the later section "Long-term cycling simulation under thermal management".

**The coupled effects of inhomogeneous current distribution and local resistance on available cell capacity**. The effect of resistance inhomogeneity on the cell's available capacity can be explored using the pseudo-end-of-life (pEoL) model developed without having to run the degradation model over many hundreds or thousands of cycles. The straightforward case of a gradient in internal layer resistance along the thickness of the cell is assumed in the pEOL model, as shown in Fig. 3(a). The modified resistance is denoted as $R_i^{EoL}$, where $i$ is the index of the electrode pair units along the cell thickness direction (i.e., $i = 1$ for the top

unit and $i = 5$ for the bottom unit, for a discretisation into 5 units along the thickness, as schematically shown in Fig. 3(a)). For the multiple units along the length and width directions, there is no imposed resistance gradient. The EoL cell-level resistance $R_{Cell}^{EoL}$ is set to twice the BoL resistance, since a cell is often considered to reach EoL when the lumped resistance is doubled, i.e.,

$$R_{Cell}^{EoL} = 2 \cdot R_{Cell}^{BoL}. \tag{16}$$

The cell resistance is related to all the ECN units connected in parallel by:

$$\frac{1}{R_{Cell}^{EoL}} = \sum_{j=1}^{nECN} \frac{1}{R_j^{EoL}}, \tag{17}$$

where $nECN$ is the number of electrical ECN units. The lumped cell resistance ($R_{Cell}^{EoL}$ in Eq. (16)) is the only resistance measurable via experimental characterisation methods, and is fixed in this study. The normalised resistance difference $\Delta\bar{R}$ describes the severity of resistance non-uniformity:

$$\Delta\bar{R} = \frac{R^{max} - R^{min}}{R^{min}}, \tag{18}$$

where $R^{max}$ and $R^{min}$ are the maximum and minimum resistance of ECN units anywhere in the cell, here along the cell thickness, as shown in Fig. 3(a). Given $\Delta\bar{R}$ in Eq. (18), combining Eqs. (16), (17) and the linear gradient assumption in Eq. (15), allows all the ECN unit resistances $R_k^{EoL}$ in the pEoL model to be calculated.

The effect of resistance inhomogeneity on the available cell capacity is explored for four magnitudes of the inner resistance gradient, $\Delta\bar{R} = 0$, 1, 5 and 10, under 6 C discharge, with a cutoff

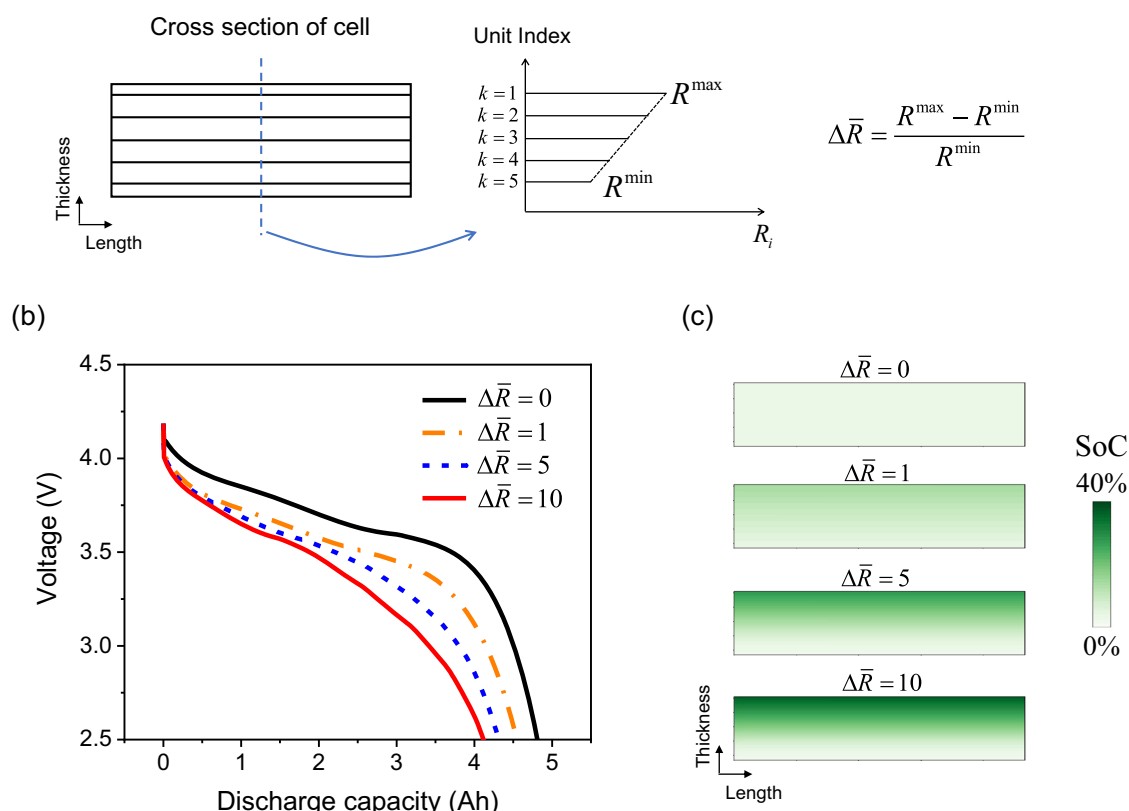

**Fig. 3 Internal resistance inhomogeneity has a strong effect on cell discharge capacity. a** Schematic of resistance distribution along the thickness direction in the pEoL model. **b** Evolution of the terminal voltage during a 6C isothermal discharge at 25 °C for normalised resistance difference of $\Delta\bar{R} = 0$, 1, 5 and 10. **c** Cross-sectional view of SoC distribution for $\Delta\bar{R} = 0$, 1, 5 and 10 at the end of the discharge.

voltage of 2.5 V and fixed 25 °C cell temperature, thus excluding the effect of self-heating.

The evolution of the terminal voltage discharge is shown in Fig. 3(b). It is apparent that both the cell terminal voltage and the cell available capacity decrease significantly as the resistance inhomogeneity (i.e., $\Delta\bar{R}$) increases, despite the lumped cell resistance $R_{Cell}^{EoL}$ having the same value. The available discharge capacity is reduced from 4.81 Ah (at homogeneous degradation given by $\Delta\bar{R} = 0$) to 4.54 Ah (at $\Delta\bar{R} = 1$), 4.31 Ah (at $\Delta\bar{R} = 5$) and 4.11 Ah (at $\Delta\bar{R} = 10$). The remaining SoC across the cell at the end of discharge is shown in Fig. 3(c). Significant remaining SoC inhomogeneity is found, demonstrating higher resistance inhomogeneity leads to larger current/SoC inhomogeneity. Towards the end of the discharge, increasingly larger currents must be provided by a small number of layers, those with the higher resistance. The measured cell resistance is thus higher compared to the no-inhomogeneity case, and the cutoff voltage is reached earlier in the discharge. The capacities corresponding to the different cell units are not evenly discharged and a larger proportion of the cell capacity remains unutilised when the cutoff voltage is reached. A reduction in useable capacity of greater than 5% occurs for $\Delta\bar{R} = 1$. Since resistance inhomogeneities of this magnitude are regularly reported[43], the failure to take this effect into account when modelling degradation, e.g., when using any lumped models, is almost certain to lead to under-estimation of the cell's degraded state, or to over-estimating the rate of the

degradation mechanism being assumed. Crucially, this effect cannot be detected from whole cell resistance measurements: $R_{Cell}^{EoL}$ is the same in all cases.

During charge and discharge, the local current is non-uniform as a result of the resistance distribution. It has been shown that degradation (including resistance increase) of electrode materials is strongly dependent on current, via physics-based modelling[7], SEI layer measurement[8] and post-mortem analyses[43]. Even at the pack level, where the cells connected in parallel are analogous to the regions within a battery cell, non-uniform current between cells is a key factor for accelerating degradation[21,44]. The effect of resistance inhomogeneity on current distribution is analysed using the pEoL model. Figure 4 shows the internal currents through the ECN units located along the central axis through the thickness of the cell. For the system with $\Delta\bar{R} = 0$, the current through the five units is similar and stable throughout the discharge process, as shown in Fig. 4(a). This behaviour is expected to continue as the cell degrades, as long as any other sources of inhomogeneity, such as caused by thermal boundary conditions, are absent. As shown in Fig. 4(b–d), current non-uniformity occurs when the normalised resistance difference $\Delta\bar{R}$ is non-zero, with the layers contributing the highest/lowest current swapping at different points during the discharge. This current swapping behaviour has been frequently reported in parallel connected small battery modules and attributed to interconnect resistance differences[21,44].

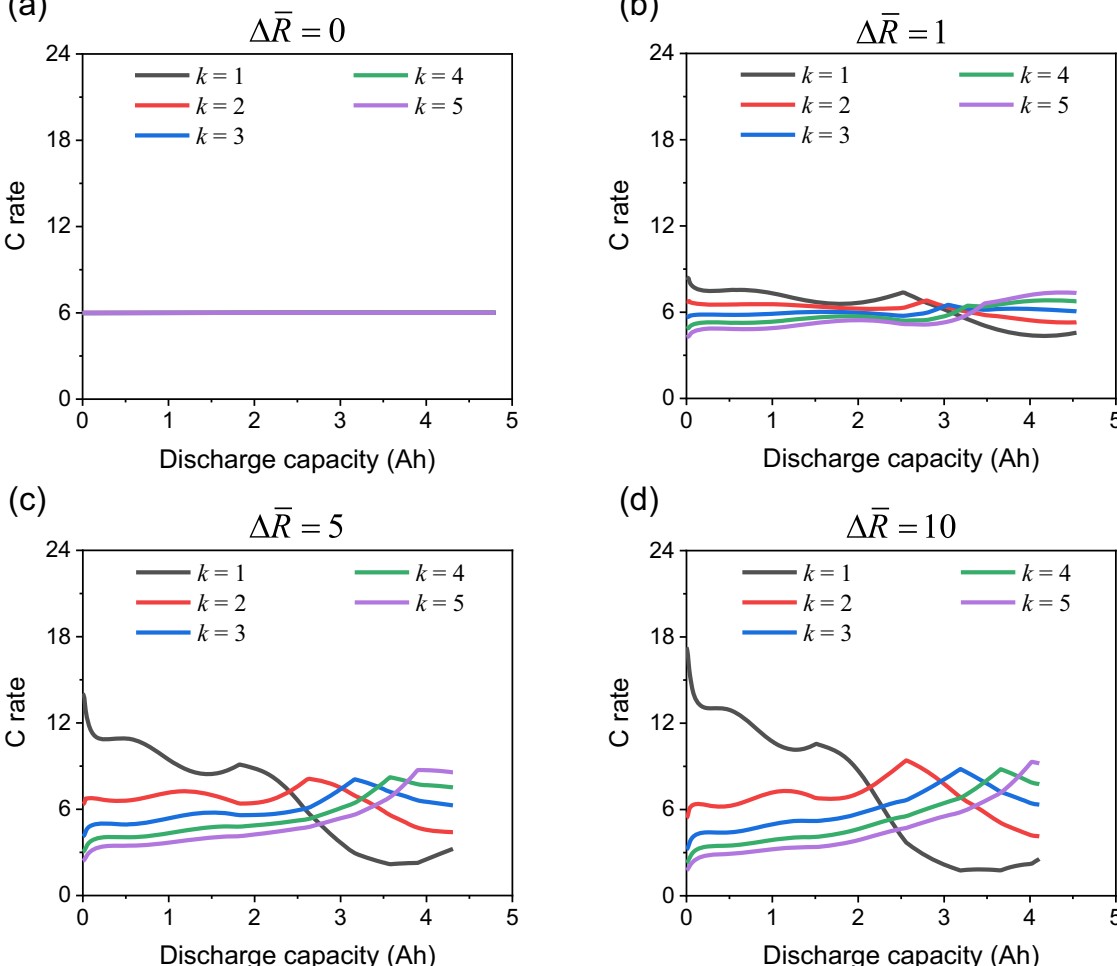

**Fig. 4 Current behaviour in a 6C discharge for the four modelled systems. a** $\Delta\bar{R} = 0$, **b** $\Delta\bar{R} = 1$, **c** $\Delta\bar{R} = 5$ and **d** $\Delta\bar{R} = 10$. The index $k$ counts along the cell thickness direction (i.e., in Fig. 3(a), $k = 1$ for the top unit and $k = 5$ for the bottom unit). The simulation conditions are the same as for Fig. 3.

The SEI growth rate has been found to be extremely sensitive to C rate in a number of studies[7,45]. A recent in-situ experimental study measured local SEI layer thickness growth rates up to ten times higher when doubling the C rate[8]. Current dependency was also found and explained by gas formation[43].

In Fig. 4(a) the current through all the five units is equal to 6 C throughout, where here the C rate is the local current rate, defined for each unit within a cell as current normalised by unit capacity. In contrast, the local currents in Fig. 4(d) vary significantly throughout the discharge, with C rates between 1.8 and 17.2 C. In the case of current-dependent degradation, accelerated degradation would be triggered in Fig. 4(d) more than in Fig. 4(a), no matter whether the ageing law increases or decreases with the current. Therefore, resistance inhomogeneity due to degradation is expected to increase faster for a cell where current inhomogeneity is more significant. Thus, a positive feedback between inhomogeneities of current and resistance within a cell is expected.

There is a distinct lack of published degradation studies that consider the effect of inhomogeneous resistance increase inside a cell. Moreover, to the best of the authors' knowledge, all existing ageing laws have been parametrised from pouch cells[42] or cylindrical cells[2,46], where uneven internal temperature can easily occur, but have been ignored. These experiments were typically performed using active surface cooling, such as occurring in thermal chambers with forced air convection, for which significant inhomogeneity of temperature, current and resistance increase is expected. To demonstrate the pitfall of ignoring inhomogeneities, Fig. 5 illustrates the simplified situation of two layers (or groups of layers) that are connected in parallel within a cell. If the resistance of one layer/group is increased by 10 times, the lumped resistance for the system only almost doubles. It is common that a cell is considered EoL when its resistance

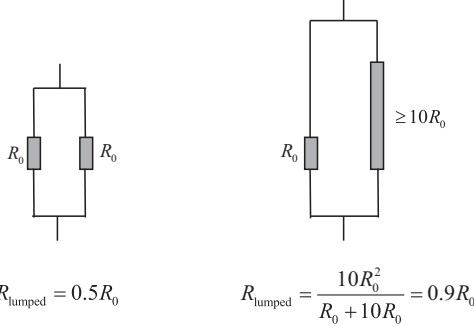

$$R_{lumped} = 0.5R_0 \qquad R_{lumped} = \frac{10R_0^2}{R_0 + 10R_0} = 0.9R_0$$

**Fig. 5 Schematic of resistance inhomogeneity and lumped resistance** $R_{lumped}$. The lumped resistance of a cell is not sensitive to resistance inhomogeneity, even when the latter is significant.

has doubled, yet this demonstrates that significant resistance inhomogeneities can occur before this limit is reached. Considering a degradation function that is strongly current and temperature-dependent, and the dependency of local current on the local resistance and temperature, we expect significant inhomogeneities to emerge in local resistance, either as a result of inhomogeneity present at the BoL or introduced through poor thermal management. A lumped model cannot recreate this dependency, making all degradation functions[2,42,46] generated from fitting lumped models to experimental data incorrect, if the data is affected by inhomogeneities.

The inhomogeneities of resistance are experimentally verified by post-mortem electrode imaging on a degraded cell after 1000 cycles. The mid-stack anode and edge anode layers of the surface-cooled cell are characterised by Scanning Electron Microscopy (SEM). The centre layer (Fig. 6(a)) is rougher than the outside layer (Fig. 6(b)) with more pitting and flakes. Using the open-source image process software Image J, the roughness Ra value is extracted as 25.05 and 20.73 (0–255 in grayscale) for centre layer and outside layer, respectively. This surface roughness difference can be interpreted as indicative of a more pronounced SEI layer in the mid-stack layer, as observed in experimental work[47].

**Choosing a degradation function with current dependency.** The choice of degradation functions is discussed below, as the details of its formulation necessarily affect the relationship between current and resistance and their inhomogeneities. There are multiple possible degradation functions that can be selected from the available literature. Considering the significant effect of current on degradation functions[7,45], C rate-dependent degradation functions are preferred. In contrast, most published physical models of SEI layer growth assume that diffusion of electrolyte species through the SEI layer is a process limited by the diffusion rate, and therefore a function of SoC[48], SEI thickness[49] and temperature-dependent diffusion rate[50,51], but independent of C rate[52–54]. Recently, Attia et al.[55] concluded that charged species migration, affected by the C rate, must be considered in addition to diffusion when fitting the experimental data. Numerous models were modified in order to reproduce a current dependence with physical meaning[28,56–58]. Cracking can expose new electrode surface area and lead to accelerated SEI layer growth, and can occur at high C rates during both charging and discharging[59]. The SEI growth rate can either decrease with C rate due to SEI film agglomerating into spheres at high C rates[25], or increase with C rate because of the gas evolution[43] or lithium ethylene dicarbonate accumulation[7]. Lithium plating was also observed to be highly dependent on C rate, with capacity fade ensuing faster at higher currents[1]. As a result of these complex mechanisms, the rate of resistance increase has been experimentally shown to both decrease[25] and increase[43] with C rate. In this work, degradation functions that mimic both

(a)  Inside layer

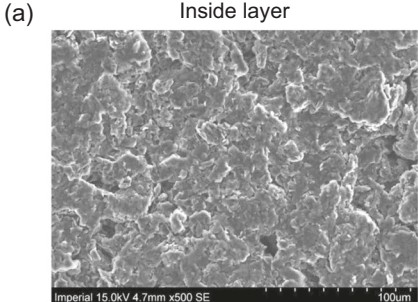

(b)  Outside layer

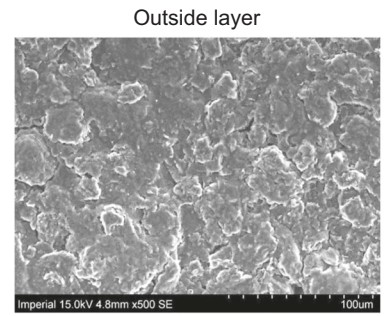

**Fig. 6 Post mortem analysis and experimental validation.** SEM image of **a** mid-stack anode layer and **b** edge anode layer from a surface-cooled Kokam cell after 1000 cycles.

scenarios were tested, i.e., higher degradation at lower C rates, and higher degradation at higher C rates, as described in detail in Supplementary Note 2.

**Long-term cycling simulation under thermal management**. Thermal management is key to the safety and longevity of the battery system. In this section, the electrical-thermal-degradation model is used to predict the effect of thermal management strategies on the cell's cycle life. The interactions between inhomogeneities of current, resistance and temperature are studied for two different cooling schemes: surface cooling (SC) and tab cooling (TC).

The current distribution inside the cell is significantly affected by the imposed thermal boundary conditions due to the strong dependence of resistance on temperature and the highly parallel electrical connections inside a cell[20]. Therefore, even at BoL current inhomogeneity can be triggered by thermal management; in fact, the more effective cooling approaches often cause higher thermal and thus current inhomogeneities[40]. To investigate how much current inhomogeneity the two thermal management approaches[19] can induce at BoL, model predictions are created for BoL during one cycle for the same protocol as used by Hunt et al.[19]: 6 C constant current voltage-limited discharge from fully charged to 2.5 V, immediately followed by 2 C constant current voltage-limited charge to 4.2 V, without constant voltage stage or any rest that could lead to internal thermal gradients dissipation.

The temperature distribution across the x-z section of the cell at its centre line is plotted in Fig. 7(a) at the end of discharge, for SC and TC. For SC the temperature inhomogeneity is significantly larger than for TC, with the temperature being highest in the core, as shown in Fig. 7(a). The impact of the cooling strategy on the temperature inhomogeneity can be understood by carefully considering the thermal conductivities of the cell components. Using the thermal properties of the pouch cell, the equivalent (lumped) in-plane thermal conductivity (i.e., along the x and y axes) can be calculated as:

$$\lambda_x^{eq} = \frac{\lambda^{Cu}\delta^{Cu} + \lambda_y^{El}\delta^{El} + \lambda^{Al}\delta^{Al} + \lambda_y^{El}\delta^{El}}{\delta^{Cu} + \delta^{El} + \delta^{Al} + \delta^{El}} = 60.50(W \cdot m^{-1}K^{-1}).$$

(19)

The equivalent through-plane thermal conductivity (i.e., along the z-axis) is given by:

$$\lambda_z^{eq} = \frac{\delta^{Cu} + \delta^{El} + \delta^{Al} + \delta^{El}}{\delta^{Cu}/\lambda^{Cu} + \delta^{El}/\lambda_z^{El} + \delta^{Al}/\lambda^{Al} + \delta^{El}/\lambda_z^{El}} = 0.91(W \cdot m^{-1}K^{-1}),$$

(20)

where $\delta^{Cu}, \delta^{El}, \delta^{Al}$ are the thicknesses of the copper foil, electrode pair (including separator) and aluminium foil, respectively, $\lambda^{Cu}, \lambda^{Al}$ are the isotropic thermal conductivities of the copper and aluminium foil, and $\lambda_x^{El}, \lambda_z^{El}$ are the thermal conductivities of

# Surface cooling

# Tab cooling

(a) Temperature $T$

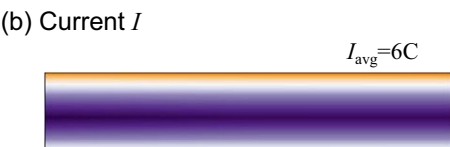

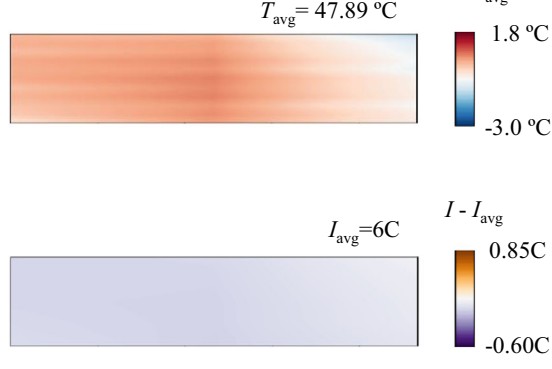

(b) Current $I$

(c)

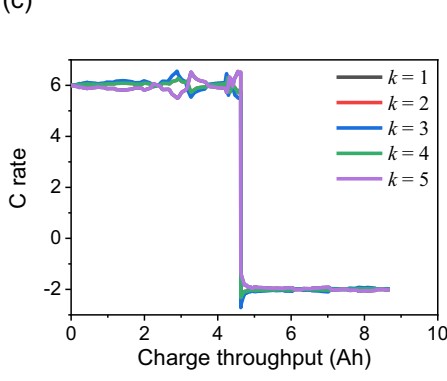

(d)

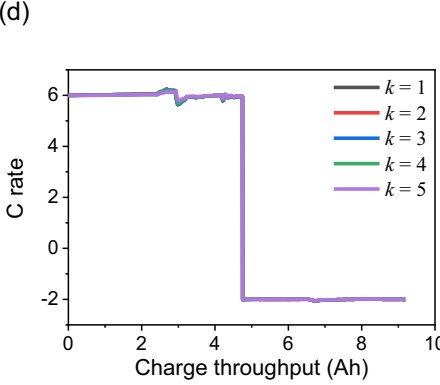

**Fig. 7 Comparison of the two thermal management strategies.** Surface cooling and tab cooling on **a** internal temperature and **b** internal current distribution at the central cross-section of the cell at the end of a 6 C discharge. Time-evolution of the local internal current for 5 locations across the thickness of the cell during a 6 C discharge and a 2 C charge for **c** surface cooling and **d** tab cooling. The index $k$ is along the cell thickness direction (i.e., in Fig. 3(a), $k = 1$ for top unit and $k = 5$ for bottom unit).

the electrode pair along the $x$ and $z$ axes. For SC, the heat transfer is limited by the lowest thermal conductivity layer, resulting in a low equivalent thermal conductivity $\lambda_z^{eq}$, while heat extraction is high due to the high surface-to-volume ratio. For TC the thermal conductivity is high, because of the good heat transfer through the highly conductive current collectors, while heat extraction is limited by the relatively small tab cross-section. As a result, a higher internal thermal gradient is established for SC than for TC. As shown in Fig. 7(a), the volume-averaged temperature $T_{avg}$ for TC is predicted by the model to be 47.89 °C, much higher than the value of 27.99 °C in SC, in agreement with the experimental data published by Hunt et al.[19].

The observed temperature inhomogeneity causes a resistance inhomogeneity, which, in turn, results in a current inhomogeneity inside the cell. Current inhomogeneities are predicted to be higher for SC than for TC at the end of the first 6 C discharge, as shown in Fig. 7(b). The local current profile across the central cross-section of the cell during the discharge and charge process is shown in Fig. 7(c) for SC and in Fig. 7(d) for TC. After a single discharge, a significant current inhomogeneity is predicted in the SC cell. Long-term cycling is expected to exacerbate the inhomogeneity of current and consequently the inhomogeneity of associated degradation, and is explored below.

Model predictions are created for 500 cycles for the same protocol as the one used by Hunt et al.[19]. In each cycle, the cell is discharged at 6 C from 4.2 V to the cutoff voltage 2.5 V, followed by charging at 2 C to the cutoff voltage 4.2 V. Each run of 500 cycles takes 5 hours on a workstation with Intel® Core™ i7-7820X Processor (3.60 GHz) with 32 G RAM. The model allows direct monitoring of the current, resistance, temperature and ageing variables (capacity loss and resistance increase) of the ECN units at a time step of 1 s.

The available capacity normalised to the capacity at BoL is shown in Fig. 8(a) for the SC and TC scenarios. As seen experimentally, SC leads to accelerated degradation compared to TC. The rate of loss of available capacity is estimated by the slope of the linear fit to the data in Fig. 8(a) and given in Fig. 8(b).

The model predictions are in agreement with the experimental data: the rate of loss of available capacity is roughly three times higher for SC than for TC, significantly reducing the cell lifetime.

In the following, the model is used to explore the mechanisms leading to the accelerated degradation. The current through five different locations within the cell is plotted in Fig. 9(a), (b) for the first 3 cycles and the last 3 cycles of the 500 cycle degradation run. The five locations lie along the centre line of the cell, where $k$ is an index along the cell thickness direction (i.e., in Fig. 3(a), $k = 1$ for top unit and $k = 5$ for the bottom unit). For SC, a non-negligible current inhomogeneity is already established in the first three cycles; during the last 3 cycles the current inhomogeneities are considerably increased, as shown in Fig. 9(a). In contrast, the currents under TC are and remain homogeneous throughout cycling, as shown in Fig. 9(b). As a result of the current and temperature inhomogeneities, the increased resistance during cycling is also more inhomogeneous for SC (Fig. 9(c)) than for TC (Fig. 9(d)). The difference between the maximum and minimum temperatures within the cell $\Delta T = T_{max} - T_{min}$ is plotted in Fig. 9(e) for SC and Fig. 9(f) for TC. Since real-time $\Delta T$ is fluctuating within each cycle, the cycle-averaged $\Delta T$ is also plotted for clarity. It is shown that $\Delta T$ is significantly higher in SC than in TC. Therefore, the higher thermal gradient introduced by SC enhances the inhomogeneities of current and resistance compared to TC.

The average temperature $T_{avg}$ is also compared for the two cooling schemes. The cycle-averaged cell temperature is shown in the lower inset of Fig. 9(e) for SC and lower inset of Fig. 9(f) for TC. The $T_{avg}$ level is around 40 °C for TC, much higher than the 28 °C for SC, due to the heat extraction being limited in TC. As TC results in a higher average temperature, ageing is more severe, since here the ageing rate is modelled to increases with temperature, as it is usually observed following the Arrhenius law[42]. The ensuing loss of fundamental cell capacity is plotted in Fig. 9(g), (h). Fundamental capacity here denotes the theoretical capacity used as reference for the cell's SoC, calculated as the capacity fade given by the chosen ageing law, subtracted from the BoL capacity (5 Ah as listed in the cell

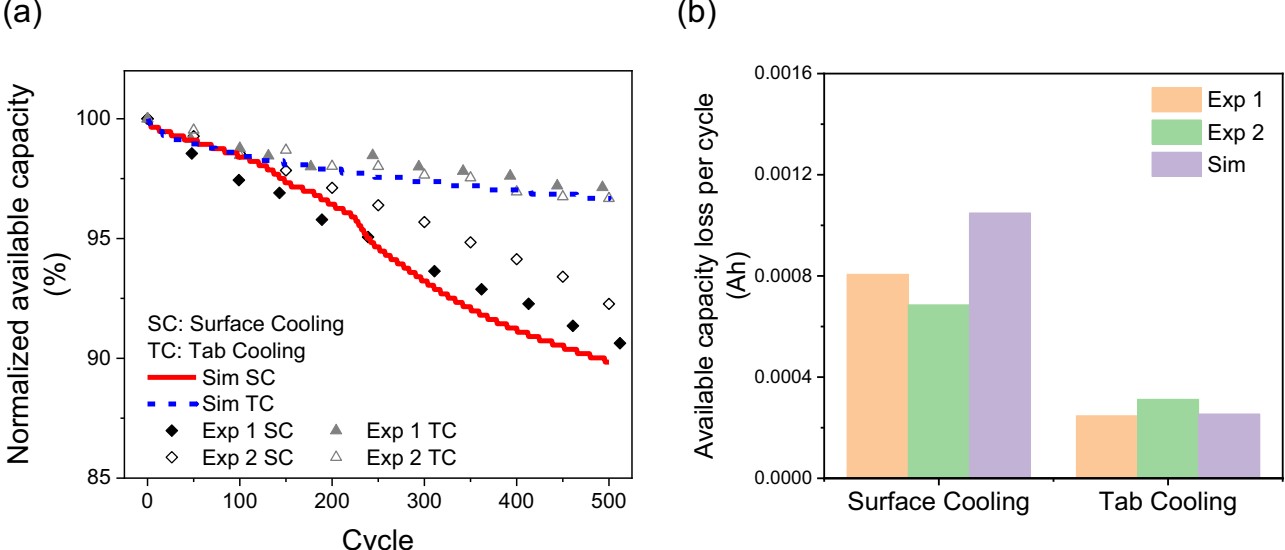

**Fig. 8 Cycling degradation results for 6 C discharge and 2 C charge for the surface cooling and tab cooling scenarios. a** Comparison of simulation results and experimental data retrieved from the study of Hunt et al.[19] for normalised available capacity. The blue dashed line and red solid line represent the simulated results under the surface cooling and tab cooling scheme, respectively. The symbols, diamonds and triangles, represent the experimental data under surface cooling and tab cooling scheme, respectively. **b** Loss of available capacity for the two cooling schemes. The orange and green bars represent the experimental data. The purple bars represent simulation results.

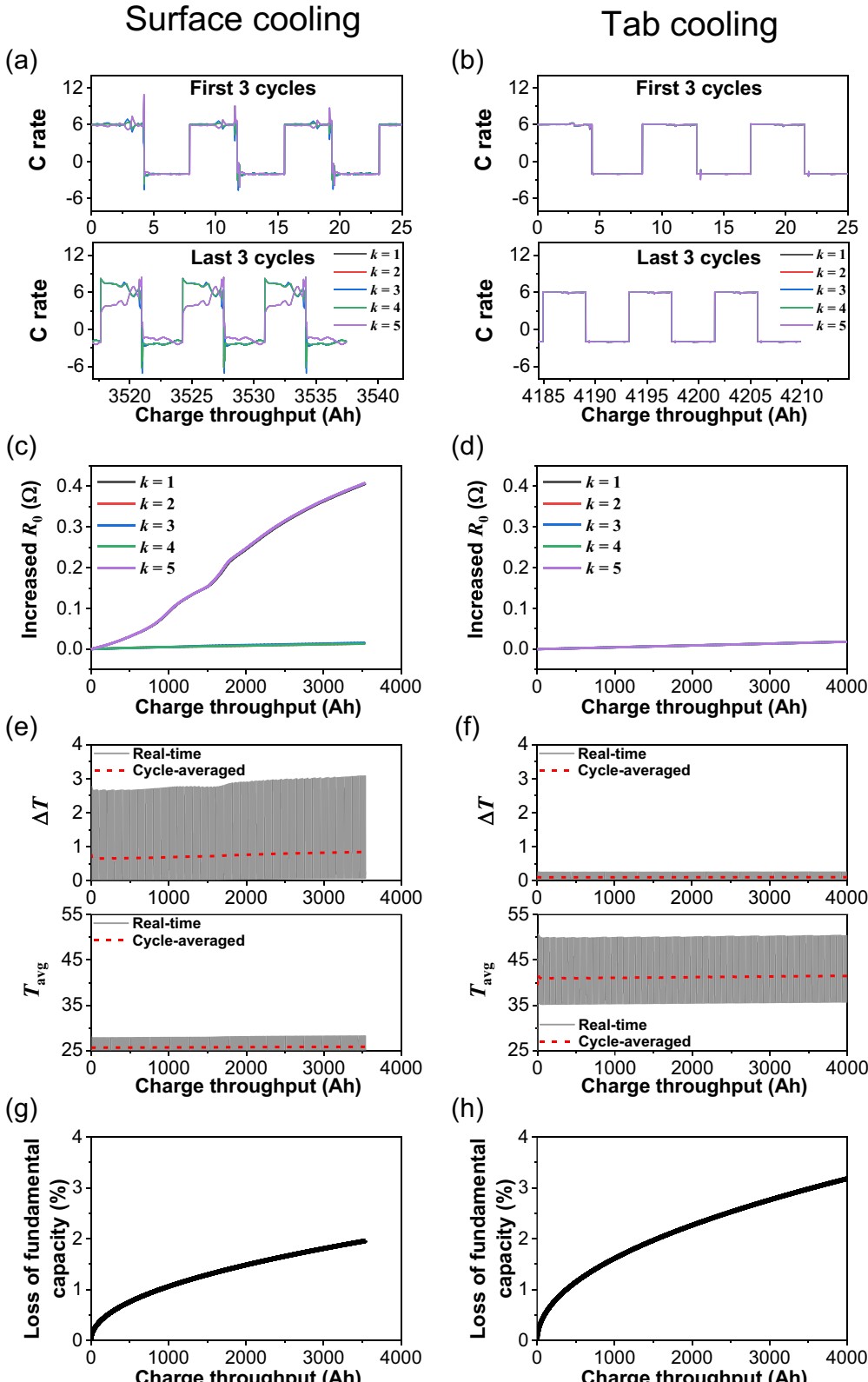

**Fig. 9 Cycle-by-cycle analysis on local states.** Local C rate distribution for the ECN units along the centre axis in the cell thickness direction for **a** surface cooling and **b** tab cooling. The upper and lower insets show the first and last three cycles, respectively, of the 500 cycles modelled. Increased resistance to $R_O$ is shown for the ECN units for **c** surface cooling and **d** tab cooling. The values for increased $R_O$ for units $k = 1$ to $k = 4$ in **c** overlap. The thermal gradient $\Delta T$ across the thickness of the cell and the average cell temperature $T_{avg}$ are shown for **e** surface cooling and **f** tab cooling. The grey lines represent the real-time data and the red dotted lines represent the cycle-averaged value. The loss of fundamental capacity percentage for the whole cell is shown for **g** surface cooling and **h** tab cooling.

data sheet). To be noted, the fundamental capacity is higher than the available, rate-dependent capacity. Loss in the fundamental capacity is one measure of ageing. Fig. 9(g) and (h) show that TC leads to nearly 50% more capacity loss over 500 cycles compared to SC. Although surface cooling facilitates lower $T_{avg}$ and thus lower loss in fundamental capacity, the larger $\Delta T$ and the ensuing gradient in degradation dominates the overall cell performance, with degradation under SC occurring considerably faster than under TC.

For a degradation function that assumes the SEI layer growth is faster at lower C rates, similar results are obtained, as shown in Supplementary Note 3. In this case, the current inhomogeneity is again initiated by the temperature gradient caused by SC. Positive feedback is initiated and results in a rapid divergence of local resistances and a rapid degradation in available cell capacity. Supplementary Note 4 presents a third simulation study that creates predictions for the case in which the current dependency of the degradation function is independent or weakly dependent on the current. It is demonstrated that, in order to retrieve the observed experimental effects of SC vs. TC, the degradation function must strongly depend on the current.

## Conclusions

Experiments have previously shown that some thermal management strategies can induce significantly faster degradation than others. The reasons for this effect could only be hypothesised, but not verified with the currently available models.

A thermally coupled and distributed model including degradation functions that are a function of both temperature and current has been proposed. Both simulation and experimental results show that thermal gradients of just 3 °C within the active region of a cell accelerate battery degradation by 300%. The simulation was done using a distributed equivalent circuit network (ECN) model, which considers inhomogeneities in local temperature and current within the cell, and their interactions with different thermal management strategies.

The model shows that, for the cell being modelled, surface cooling causes significant temperature gradients, which initiate current inhomogeneity between the layers in the thickness direction of the pouch. This causes unequal degradation, as the layers degrade at different rates, which leads to permanent inhomogeneities in the fundamental resistance of individual layers. As a verification of inhomogeneities, post-mortem analysis shows a roughness difference of 20.8% between the cell inside and outside layer. As a result, there is a substantial reduction in useable capacity and energy that cannot be identified by simply measuring the lumped resistance of the cell. Indeed, we show that two cells with identical lumped resistances could have very different patterns of localised degradation that could result in substantial differences in useable capacity and energy. These effects can only be reproduced by a thermally coupled and distributed model. The model showed that for tab cooling the inhomogeneities and therefore positive feedback had a negligible impact on degradation.

The model also reproduced the experimentally observed capacity loss under slow discharge being higher for tab cooling than for surface cooling and shows it to be caused by the higher average cell temperature when tab cooling. However this degradation occurred homogeneously across the thickness of the cell, and therefore had only a moderate effect on the useable cell capacity.

It was only possible to reproduce the experimentally observed effect of inhomogeneity on degradation when a degradation function strongly dependent on current was used. This relationship was necessary to initiate the positive feedback leading to faster degradation caused by thermal gradients in the active region of the cell. Many degradation mechanisms, such as particle cracking, lithium plating and SEI layer growth, are known to be functions of current, suggesting inhomogeneity effects cannot and should not be ignored. Therefore, most attempts to reproduce realistic cell-level degradation based upon a lumped model (i.e., lacking thermal gradients) have suffered from significant overfitting, incorrectly concluding the rate of degradation of the mechanisms considered.

## Data availability

The data that support the findings of this study are available from the corresponding author upon reasonable request.

## Code availability

The code that supports the findings of this study is available from the corresponding author upon reasonable request.

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

## Acknowledgements

The authors would like to acknowledge the support from EPSRC Faraday Institution Multi-Scale Modelling project (EP/S003053/1, grant number FIRG003), Innovate UK BATMAN project (grant number 104180) and Innovate UK WIZer project (grant number 104427). The authors also acknowledge the contribution of Mr. Amir Amiri in creating the image in Fig. 1.

## Author contributions

S.L.: conceptualisation, data curation, formal analysis, investigation, methodology, software, validation, visualisation, writing—original draft, writing—review & editing. M.C.Z.: parametrisation, validation, methodology, discussion, writing—review and editing, Y.Z.: parametrisation, experiments, methodology, discussion, G.J.O.: formal analysis, funding acquisition, supervision, writing—original draft, writing—review & editing, M.M.: conceptualisation, formal analysis, funding acquisition, methodology, supervision, writing—original draft, writing—review & editing.

## Competing interests

The authors declare no competing interests.
