## [Peer Review File · Communications Engineering]

Reviewers' comments:

Reviewer #1 (Remarks to the Author):

This work established an inhomogeneous degradation for lithium-ion batteries by combining a distributed electric circuit network with an empirical ageing sub-model, which successfully explained the accelerated LIB degradation caused by temperature gradient under surface cooling. The manuscript is well organized, and the topic and findings of the paper are very interesting. I think the work is of interest to the journal and only a minor revision is requested. Here are my comments:

1. There are a few typos (like lines 210-211 and line 342).
2. The model has 5 elements along the z axis, does this physically represent the number of layers in the punch cell? If not, please explain why more elements are used along the z axis.
3. In Fig. 2, only the temperature rise during the two cooling scenarios is validated. I think it would be more convincing if the simulated voltage is also compared to the experimental results.
4. Since the degradation is purely attributed to SEI growth, is the resistance increase in Eq. 12-14 only affecting the ohmic resistance? If not, please clarify how is resistance rise distributed to different resistances of the model. In addition, please also explain whether the increased part of the degraded resistance depends on SOC and temperature like the BOL model parameters.

Reviewer #2 (Remarks to the Author):

This work has shown some interesting work in the investigation of the effects of thermal gradients on inhomogeneous aging of lithium-ion batteries. I agree with the publication after the revision considering the following points:

1. The authors have claimed that this work proposes for the first time a distributed electrical-thermal-degradation ECN model. Please be careful while using "first time". Schmalstieg has developed a model with the similar idea in this phd thesis in 2017, DOI: 10.18154/RWTH-2017-04693.
2. How was the electro-thermal ECN model parameterized? Has the model be validated only under one load profile and one ambient temperature? It is nessesary to validate this model under different working conditions, which is the difficulty of all these kind of models as we know.
3. How did you parameterize the degradation model? I mean the hyper parameters of the degradation models here. Using literature values or fitted from the experimental data?
4. Can you show the experimental validation of the work in Section 3? If it is pure simulation work without experimental validation and post mortem analysis by opening the battery cells, it is difficult to draw the conclusions.

Reviewer #3 (Remarks to the Author):

1. The literature survey is too simple without considering some types of battery electric-thermal-aging

model. Besides, using coupled battery models to analyze the effect of temperature and current inhomogeneities on the battery life have been proposed by many researchers. So the real novelty of this article should be highlighted to tell the difference of your work. Similar idea has been reported in previous papers, e.g.,

a) Physics based modeling of a series parallel battery pack for asymmetry analysis, predictive control and life extension. *Journal of Power Source*, 322(2016): 57-67.

b) A reliability design method for a lithium-ion battery pack considering the thermal disequilibrium in electric vehicles. *Journal of Power Source*, 386 (2018) 10-20.

c) Multiphysical modeling for life analysis of lithium-ion battery pack in electric vehicles. *Renewable and Sustainable Energy Reviews*, 131(2020): 109993.

This is the biggest concern from my side.

2. In the Introduction, the author said that the empirical models or semi-empirical models have been oversimplified and do not accurately describe the cycle by cycle evolution and interaction between thermal, electrical and degradation inhomogeneities. Why the empirical aging model is still used in this work? Please explain.

3. What method is used for simulation? Is it finite element method? If yes, please provide the CAD and FEA models and parameters of lithium-ion batteries.

4. Figure 6 shows that the battery temperature is relatively high under tab cooling conditions, but the capacity degradation rate in Figure 7 is relatively slow, which is inconsistent with the conclusion and common sense. Please explain the reason.

5. The benefits of the presented work is not evident, since the results are intuitive, even without the simulation. It is suggested that some quantitative conclusions should be given.

6. Some format problems need to be corrected, e.g., multiple occurrences of two spaces between words.

COMMSENG-22-0338

Title: Inhomogeneous degradation in lithium-ion batteries: the effect of thermal gradients

We thank the editor for consideration of our work and the reviewers for their careful reading and efforts in helping us to improve the manuscript. Based on the comments and suggestions from the reviewers, we have revised the manuscript accordingly. In the updated manuscript, we have added simulation results, SEM experimental results and quantitative descriptions.

Reviewer #1:

This work established an inhomogeneous degradation for lithium-ion batteries by combining a distributed electric circuit network with an empirical ageing sub-model, which successfully explained the accelerated LIB degradation caused by temperature gradient under surface cooling. The manuscript is well organized, and the topic and findings of the paper are very interesting. I think the work is of interest to the journal and only a minor revision is requested. Here are my comments:

- We thank the reviewer for the time and effort in reading our revised manuscript and providing feedback. We are grateful to the reviewer for his/her recommendation for publication in *Nature Communication Engineering*. Based on the comments, we have revised and improved the manuscript accordingly.
1. *There are a few typos (like lines 210-211 and line 342).*
 - We thank the reviewer for the careful reading. We have checked the manuscript and corrected the typos.
 2. *The model has 5 elements along the z axis, does this physically represent the number of layers in the punch cell? If not, please explain why more elements are used along the z axis.*
 - There are 100 electrode pair layers for this Kokam cell, as found out in the disassembly test [Zhao et al., J. Electrochem. Soc., 165 (2018) A3169-A3178]. For computational efficiency, 5 units along z axis are chosen with each ECN unit along z axis representing 20 layers. For convergence check, a dense mesh of 9 x 9 x 17 units was conducted with only 0.6% error to ensure the suitability of this level of discretization. The details can be found in Supplementary Note 1.

Fig. S1. Comparison of cell core temperature for distributed electro-thermal ECN model with the normal mesh and denser mesh.

We have modified the main manuscript (in Section 2.1) to include this clarification:

“There are 100 electrode pair layers as measured in the disassembly test ¹⁷. For computational efficiency, 5 units along z axis are chosen, with each ECN unit along z axis representing 20 layers. A convergence check was conducted to verify the accuracy of this level of discretization.”

3. *In Fig. 2, only the temperature rise during the two cooling scenarios is validated. I think it would be more convincing if the simulated voltage is also compared to the experimental results.*

➔ As suggested, we have added the validation results for voltage in Fig. 2.

“Under 6C constant current discharge, the simulated voltage matches the experimental results, with RMSE of 184.71 mV (for SC) and 150.82 mV (for TC), as shown in Fig. 2(e) and (f).”

Fig. 1. Electro-thermal model validation. Under adiabatic condition: (a) drive cycle voltage comparison, (b) surface temperature comparison. Under 6C constant current discharge, temperature at the centre of the measured surface for (c) single-sided surface cooling and (d) tab cooling; cell voltage for (e) single-sided surface cooling and (f) tab cooling. Experimental data is obtained from previous work.¹⁷

4. *Since the degradation is purely attributed to SEI growth, is the resistance increase in Eq. 12-14 only affecting the ohmic resistance? If not, please clarify how is resistance rise distributed to different resistances of the model. In addition, please also explain whether the increased part of the degraded resistance depends on SOC and temperature like the BOL model parameters.*

→ The resistance increase affects ECN unit resistances R_0 and R_1 to R_3 . Each resistance (R_0, R_1 to R_3) in the ECN unit is composed of two parts as shown in Eq. (14): one term giving the BoL resistance and one term giving the extra resistance due to degradation. The BoL resistance terms are a function of SoC and temperature. The resistance rise term is scaled from the BoL resistance term by the percentage μ in an incremental way as expressed in Eq. (13). Therefore, resistance rise is implicitly dependent on SoC and temperature (through its dependence on BoL resistance) and C rate (from its

dependence on the percentage μ in Eq. (11)).

The following explanation has been added to the manuscript in Section 2.3:

“The total resistance percentage increase relative to the BoL total resistance for the cell is expressed as in Cordoba-Arenas et al.³⁹ relative to the total charge throughput through the cell W :

$$\mu = \{3205.3 + 36.34 \cdot \exp[0.92(5 - CR)]\} \cdot \exp\left(\frac{-51800}{8.31T}\right) \cdot W, \quad (1)$$

where CR is the C rate at which the cell is cycled. The resistance percentage increase per volumetric density of charge throughput (w) in the i -th ECN unit corresponds to:

$$\frac{d\mu_i}{dw} = \{3205.30 + 36.34 \cdot \exp[0.92(5 - CR)]\} \cdot \exp\left(\frac{-51800}{8.32T}\right) \cdot V_0. \quad (2)$$

It is assumed that an increase in local total resistance as a result of degradation occurs as a result of proportional increases in each of the four resistances of the ECN unit. A degradation-caused increase dR_i for a resistance in the i -th ECN unit at time t is:

$$dR_i(t) = R_i^{BoL}(SoC_i(t), T_i(t)) \cdot \frac{1}{100} d\mu_i(t), \quad (3)$$

where $SoC_i(t)$ and $T_i(t)$ are the state of charge and temperature for the i -th ECN unit at time t . Thus the resistance increase dR_i is affected by SoC and temperature indirectly, through $R_i^{BoL}(t)$, and by C rate directly through $d\mu_i(t)$. Any of the four resistances in the i -th unit ECN R_i are assumed to be composed of BoL resistance plus a history-dependent increase, expressed as:

$$R_i(t) = R_i^{BoL}(t) + \int_{\tau=0}^t dR_i^{BoL}(\tau). \quad (4)$$

..

Reviewer #2:

This work has shown some interesting work in the investigation of the effects of thermal gradients on inhomogeneous aging of lithium-ion batteries. I agree with the publication after the revision considering the following points:

→ We thank the reviewer for the time and effort in reading our manuscript and providing feedback to improve the quality of the manuscript. We are grateful to the reviewer for his/her recommendation for publication in *Nature Communications Engineering*. Based on the suggestions and comments, we have corrected or rephrased the main text accordingly. We hope the reviewer will find it more clearly presented now.

1. *The authors have claimed that this work proposes for the first time a distributed electrical-thermal-degradation ECN model. Please be careful while using "first time". Schmalstieg has developed a model with the similar idea in this phd thesis in 2017, DOI: 10.18154/RWTH-2017-04693.*

→ We thank the reviewer for the comment. We have removed the word “first

time” from the manuscript. The model in Schmalstieg’s thesis [DOI: 10.18154/RWTH-2017-04693] assumes a fixed temperature (i.e., no thermal model or coupling among electrical-thermal-degradation models). The proposed model considers a 3D thermal model with different internal thermal properties (e.g., the thermal conductivity of current collector is orders higher than electrode pair) and the effect of thermal boundary condition.

2. *How was the electro-thermal ECN model parameterized? Has the model be validated only under one load profile and one ambient temperature? It is necessary to validate this model under different working conditions, which is the difficulty of all these kind of models as we know.*

→ We thank the reviewer for the comment. The electrical parameters were extracted using Layered method [Jackey et al., SAE Technical Papers, (2013), DOI: 10.4271/2013-01-1547], from experimental data of Galvanostatic Intermittent Titration Technique (GITT) tests. The thermal properties were taken from literature [Zhao et al., J. Electrochem. Soc., 165, A1846-A1852, (2018), DOI: 10.1149/2.0901813jes]. We have revised Section 2.1 as below.

“The parameters of the electrical model for the whole cell are formulated as lookup tables at different temperatures (10 °C, 20 °C, 30 °C and 40 °C) and different SoCs (from 1% to 100%). The electrical parameters (open circuit voltage E_s , resistance R_0 , R_i and capacitance C_i) are extracted using the Layered method³⁸ on experimental data from Galvanostatic Intermittent Titration Technique (GITT) tests conducted in discharge at 1C (5 A). The electrical parameters and thermal properties (thermal conductivity λ , specific heat capacity c and mass density ρ for aluminium foil, copper foil, anode, cathode and separator) taken from previous work on the same cell.¹⁷”

The electro-thermal model is validated against drive cycle under adiabatic condition and 6C constant current discharge condition under surface cooling and tab cooling approach, as shown in Fig. 2 in the manuscript.

3. *How did you parameterize the degradation model? I mean the hyper parameters of the degradation models here. Using literature values or fitted from the experimental data?*

→ The ageing law for resistance increase is taken from literature where degradation is comprehensively parametrized under different temperature and current [Cordoba-Arenas et al., J. Power Sources, 2014, DOI: 10.1016/j.jpowsour.2014.12.047]. The details of the ageing function used in the model are described in Supplementary Note 2.

4. *Can you show the experimental validation of the work in Section 3? If it is pure simulation work without experimental validation and post mortem analysis by opening the battery cells, it is difficult to draw the conclusions.*

→ We thank the reviewer for the constructive comment. In the revised manuscript, we have added the post mortem analysis (SEM image roughness analysis) as evidence for inhomogeneous degradation on the surface-cooled cell. The following discussion and images have been added to Section 3.1:

“The inhomogeneities of resistance are experimentally verified by post-mortem electrode imaging on a degraded cell after 1000 cycles. The mid-stack anode and edge anode layers of the surface-cooled cell are characterised by Scanning Electron Microscopy (SEM). As shown in Fig. 6, the center layer is rougher than the outside layer with more pitting and flakes. Using the open-source image process software Image J, the roughness Ra value is extracted as 25.05 and 20.73 (0 – 255 in grayscale) for center layer and out-side layer, respectively. This surface roughness difference can be interpreted as indicative of a more pronounced SEI layer in the mid-stack layer, as observed in experimental work⁴⁴.”

Fig. 6. Post mortem analysis and experimental validation. SEM image of (a) mid-stack anode layer and (b) edge anode layer from a surface-cooled Kokam cell after 1000 cycles.

Reviewer #3:

1. *The literature survey is too simple without considering some types of battery electric-thermal-aging model. Besides, using coupled battery models to analyze the effect of temperature and current inhomogeneities on the battery life have been proposed by many researchers. So the real novelty of this article should be highlighted to tell the difference of your work. Similar idea has been reported in previous papers, e.g.,*

a) Physics based modeling of a series parallel battery pack for asymmetry analysis, predictive control and life extension. Journal of Power Source, 322(2016): 57-67.

b) A reliability design method for a lithium-ion battery pack considering the thermal disequilibrium in electric vehicles. Journal of Power Source, 386 (2018) 10-20.

c) Multiphysical modeling for life analysis of lithium-ion battery pack in electric vehicles. Renewable and Sustainable Energy Reviews, 131(2020): 109993.

This is the biggest concern from my side.

→ We thank the reviewer for the comment. The novelty of our work is three fold:

- i. We created a 3D cell-level electrical-thermal-degradation model that captures the effect of thermal management and consequent inhomogeneities of current and resistance inside a single battery cell. We show that the internal isotropic structure of the cell is determinant in enabling inhomogeneous degradation. The model is suitable for long-term degradation simulation and the model is validated against long-term cycling experiments.
- ii. The mechanism of the accelerated degradation of surface cooled pouch cells compared to tab cooled pouch cells that was found experimentally is explained by the model predictions as being the result of the interaction of current and degradation inhomogeneities (see Fig. 9 in our manuscript).
- iii. We conclude that the majority of existing cell level modelling approaches based on lumped models are probably wrong in their prediction of degradation, which is expected to be non-homogeneous, especially for realistic applications in which the cell is subjected to highly inhomogeneous boundary conditions.

Ganesan's work [Ganesan et al., J. Power Sources, 2016] creates an electrical-thermal-degradation model for pack level. The model considers thermal gradient, current inhomogeneity and degradation inhomogeneity on battery pack using lumped cell model. Our work is focused on cell-level distributed models.

Besides, Ganesan's work discussed the simulated capacity fade in a predictive way. There is no validation on long-term cycling degradation or analysis on the accelerated degradation. In our work, the model is validated against long-term cycling degradation experiments. We then explain the emergence and propagation of mechanisms of accelerated degradation by analyzing the interaction of current and degradation inhomogeneities (see Fig. 9 in our manuscript) during the full ageing history.

Xia's work [Xia et al., J. Power Sources, 2018] also discussed the pack level model. The degradation is pre-assumed in a stochastic way. Another work [Xia et al., Renewable and Sustainable Energy Reviews, 2020] also focuses on pack level model. The interaction between the temperature, current and resistance is not analysed.

We have added those literature in Introduction section.

“Thermal management was found to play an important role in slowing down degradation processes for single cell and battery pack due to inhomogeneities¹⁶⁻¹⁸.”

2. *In the Introduction, the author said that the empirical models or semi-empirical models have been oversimplified and do not accurately describe the cycle by cycle evolution and interaction between thermal, electrical and degradation inhomogeneities. Why the empirical aging model is still used in this work? Please explain.*

→ We thank the reviewer for the comment. We are sorry to cause the confusion in using the word ‘empirical model’. The proposed electrical-thermal-degradation model is composed of distributed electro-thermal ECN model and degradation functions which are derived in an incremental way to enable cycle-by-cycle evolution. The previous simplified empirical models are not capable of cycle-by-cycle evolution analysis.

We have revised the Introduction section with the sentences below to clarify this point.

“This work proposes a novel distributed electrical-thermal-degradation equivalent circuit network (ECN) model with the ability to track the cell's local states as they evolve cycle-by-cycle. A degradation function for capacity and power fade is adapted from previously published studies and used within the 3D distributed electro-thermal model to progress degradation, in the form of increased resistance, incrementally and locally within the cell. This allows the model to retrieve the interactions between the inhomogeneities in temperature, resistance, current, state of health and rate of degradation.”

3. *What method is used for simulation? Is it finite element method? If yes, please*

provide the CAD and FEA models and parameters of lithium-ion batteries.

- The 3D electro-thermal model is developed using Finite Difference Method. The schematic of the model is shown in Fig. 1. We have updated the manuscript with the description of model parameters used. We hope the reader will find them more clearly than in previous version.

“An electro-thermal model for a pouch cell was developed using the Finite Difference Method, as previously used in the creation of a 3D cylindrical cell model³⁷ and a 2D pouch cell model¹⁷.”

“The parameters of the electrical model for the whole cell are formulated as lookup tables at different temperatures (10 °C, 20 °C, 30 °C and 40 °C) and different SoCs (from 1% to 100%). The electrical parameters (open circuit voltage E_s , resistance R_0 , R_j and capacitance C_j) are extracted using the Layered method³⁸ on experimental data from Galvanostatic Intermittent Titration Technique (GITT) tests conducted in discharge at 1C (5 A). The electrical parameters and thermal properties (thermal conductivity λ , specific heat capacity c and mass density ρ for aluminium foil, copper foil, anode, cathode and separator) taken from previous work on the same cell.¹⁷”

4. *Figure 6 shows that the battery temperature is relatively high under tab cooling conditions, but the capacity degradation rate in Figure 7 is relatively slow, which is inconsistent with the conclusion and common sense. Please explain the reason.*

- The common sense is that average temperature dominates the capacity fade, i.e., higher temperature causes faster degradation rate.

However, here we show that degradation inhomogeneity dominates the degradation rate when thermal gradients are obvious, in contrast with the common sense. The previous experimental work [Hunt et al., J. Electrochem. Soc., 2016, DOI: 10.1149/2.0361609jes] has shown that surface cooling causes faster degradation than tab cooling even though the average temperature is much lower in surface cooling.

Section 3 discussed the effect of inhomogeneity on capacity fade: uneven resistance and current interact with each other in lowering down the capacity. Fig. 9 in Section 4 analyzed the history of internal resistance, current, thermal gradient and average temperature for the surface cooling and tab cooling results. It is concluded that the interaction of resistance and current inhomogeneities under surface cooling are responsible for the accelerated degradation.

5. *The benefits of the presented work is not evident, since the results are intuitive, even without the simulation. It is suggested that some quantitative conclusions should be given.*

→ We thank the reviewer for the comment. The experimental results of accelerated degradation by 300% dominated by thermal gradients and inhomogeneities have been acknowledged by the community in the work [Hunt et al., J. Electrochem. Soc., 2016, DOI: 10.1149/2.0361609jes] and [Zhao et al., J. Electrochem. Soc., 2018, DOI: 10.1149/2.0901813jes]. The simulation work here quantitatively reproduced those experimental results and revealed the mechanism of the accelerated degradation of surface cooled vs tab cooled pouch cells: the result of the interaction of current and degradation inhomogeneities. The post mortem analysis (SEM image roughness analysis in Fig.6) were also performed to verify the existence of degradation inhomogeneity. We believe our results are not intuitive.

We have revised the Conclusion by adding quantitative findings and simulation details.

“Both simulation and experimental results show that thermal gradients of just 3 °C within the active region of a cell accelerate battery degradation by 300%. The simulation was done using a distributed equivalent circuit network (ECN) model, which considers in-homogeneities in local temperature and current within the cell, and their interactions with different thermal management strategies.”

“As a verification of inhomogeneities, post mortem analysis shows a roughness difference of 20.8% between cell inside and outside layer.”

6. *Some format problems need to be corrected, e.g., multiple occurrences of two spaces between words.*

→ We thank the reviewer for checking the manuscript format. We have carefully checked and corrected the manuscript again.

REVIEWERS' COMMENTS:

Reviewer #1 (Remarks to the Author):

The authors have addressed my comments. I think the manuscript can be published in its current form.

Reviewer #2 (Remarks to the Author):

The authors have revised the paper based on my comments. I think the current version of the paper can be published.

Reviewer #3 (Remarks to the Author):

All my concerns have been addressed.